

# Assessing soil compaction and micro-topography impacts of alternative heather cutting as compared to burning as part of grouse moor management on blanket bog

Andreas Heinemeyer[1], Rebecca Berry[2,3] and Thomas J. Sloan[2]

[1] Stockholm Environment Institute at the Department of Environment and Geography, University of York, York, United Kingdom

[2] Department of Environment and Geography, University of York, York, United Kingdom

[3] Current affiliation: Bibury Road, Aldsworth, Cheltenham, Gloucestershire, UK

Corresponding author
Andreas Heinemeyer,
andreas.heinemeyer@york.ac.uk,
ah126@york.ac.uk

## ABSTRACT

**Background**. Over 25% of the UK land area is covered by uplands, the bulk of which are comprised of blanket bog. This not only contains most of the UK's terrestrial carbon stocks, but also represents 15% of this globally rare habitat. About 30% of UK blanket bog is managed for red grouse by encouraging ling heather (*Calluna vulgaris*) with rotational burning, which has been linked to habitat degradation, with reduced carbon storage and negative impacts on water storage and quality. Alternative cutting is currently being pursued as a potential restoration management. However, the often used heavy cutting machinery could cause considerable compaction and damage to the peat surface. Two particular issues are (i) a potential increase in bulk density reducing water storage capacity (i.e., less pore volume and peat depth), and (ii) a possible reduction of the micro-topography due to cutting off the tops of hummocks (i.e., protruding clumps or tussocks of sedges).

**Methods**. We set up a fully replicated field experiment assessing cutting versus burn management impacts on peat physical and surface properties. Both managements reflected commonly used grouse moor management practice with cutting using heavy tractors fitted with load distributing double wheel and tracks (lowering ground pressure), whilst burning was done manually (setting heather areas alight with flame torches). We assessed management impacts on peat depth, bulk density and peat surface micro-topography which either included pre-management measurements or plot-level data for uncut plots. Total peat depth and bulk density in four 5 cm sections within the top 50 cm was assessed. Micro-topography was determined as the standard deviation of the height offsets measured over several plot transects in relation to the plot peat surface level at the start and end points of each transect.

**Results**. Despite an anticipated compaction from the heavy machinery used for cutting, the peat showed resilience and there was no lasting plot-level impact on either peat depth or bulk density. Notably, bulk density showed differences prior to, and thus unrelated to, management, and an overall increasing bulk density, even in uncut plots. However, cutting did reduce the plot micro-topography by about 2 cm, mostly due to removing the tops of hummocks, whereas burnt plots did not differ from uncut plots.

**Discussion**. Cutting is suggested as a suitable alternative to burning on grouse moors, although compaction issues might be site specific, depending on the nature of the peat, the machinery used and impacts at resting and turning points (which were not assessed). However, any observed bulk density differences could reflect natural changes in relation to changes in peat moisture, requiring adequate experimental comparisons. Moreover, where micro-topography is a priority, cutting equipment might need to consider the specific ground conditions, which could involve adjusting cutting height and the type of cutting machinery used.

## INTRODUCTION

Uplands cover over 25% of the UK (*Haines-Young et al., 2000*). The bulk of this area comprises blanket bog, dwarf-shrub heath and acid grassland. The term "blanket bog" is often used rather loosely and falls within the overarching term of wetland, where conditions are such that soil water-logging favours *Sphagnum* moss growth and peat formation such that bog development covers all but the more steeply sloping ground with an average thickness of 0.5–3.0 m (*JNCC, 1999*). Moreover, the terms "mire" and "bog" are often seen as inter-changeable within the context of UK blanket bog, yet are in fact complex ecosystems with quite different hydrology, soil structure and plant species composition (*Lindsay et al., 1988*; *O'Brien, Labadz & Butcher, 2007*). Importantly, the UK contains about 15% of the global blanket bog areas (*Tallis, 1998*; *Evans, Warburton & Yang, 2006*), of which upland bog represents about 90% of the UK's peatland area, containing an estimated 2,300 Mt of soil organic carbon (SOC) (*Billett et al., 2010*) whilst also providing a diverse habitat for many upland plant and bird species (*Bain et al., 2011*). Crucially, the habitat diversity, including its micro-topography (i.e., hummocks and hollows), are important in supporting a range of specialist species including many in the UK rare upland birds (e.g., *Stroud et al., 1988*). Peatlands (including the habitats bogs and dwarf shrub and heath) deliver a wide range of ecosystem services (*Haines-Young & Potschin, 2008*) that contribute to human well-being, including climate regulation, water purification, maintenance of biodiversity, recreational and educational opportunities, and tourism (*Kimmel & Mander, 2010*). However, many upland bog areas in the UK are managed for either agriculture or sporting purposes (see chapter 5 in *Bain et al., 2011*) with some reported negative impacts on ecosystem services (*O'Brien, Labadz & Butcher, 2007*). Therefore, any detrimental management impacts on bog ecosystem functioning need to be assessed in order to advise on best practice management towards ecosystem services provisioning.

In the UK, grouse shooting estates (i.e., red grouse; *Lagopus lagopus scotica*) cover an estimated area of 5–15%, or between 0.66 to 1.7 million hectares of upland areas (*Grant et al., 2012*). Burning has been used to manage upland vegetation in Britain for centuries and prescribed rotational burning of ling heather (*Calluna vulgaris*) has been

used to maximise grouse densities by preventing the establishment of woody species and encouraging nutrient cycling, thus stimulating *Calluna* growth and dominance (*Yallop, Clutterbuck & Thacker, 2009*). This management creates a mosaic of different aged stands to provide both food from new young growth and protection from predators in the taller, older stands. In the UK, about 18% of peatlands (*Worrall et al., 2010*) and 30% of blanket bog (*Natural England, 2010*) are estimated to be under such a burn rotation. Although burning mostly occurs on shallow peat and podzolic soils, it regularly encroaches on deeper blanket bog peat areas (*Yallop et al., 2005*), and often peat depth is unknown to, or not necessarily taken into account by, the land manager. Large scale and repeated heather burn rotations (typically 10% of the estate's total heather area in patches of about 0.5–1.0 ha in size on a 10–15 year rotation) have only been introduced across the UK during the past 200 years (e.g., *Hay, 2012*) in relation to grouse management. However, burn rotations vary regionally and shorter or longer rotations have been noted (*Yallop et al., 2006*). Such burn rotation management encourages different age structures of heather to support high grouse populations, but when done poorly or when fires get out of control, it can also damage the peat and or kill bog mosses mainly responsible for maintaining peat hydrological function (i.e., high water holding capacity increasing water tables and soil moisture) and therefore threaten the continued formation of peat (*Natural England, 2010*).

As burning (largely in connection with drainage) has been considered to have contributed to blanket bog degradation (*Natural England, 2010*), alternative management is of interest to policy makers and land managers. One aspect of alternative management is reprofiling and blocking drainage ditches to raise water tables and encourage 'active' bog vegetation. Moreover, alternative heather management methods could also be used to reduce 'over'-domination of heather in favour of more diverse and 'active' peatland forming vegetation, especially with regard to the restoration of peat hydrological function and *Sphagnum* growth. *Sphagnum* is recognised as a unique global carbon store, containing more carbon than any other plant genus (*Clymo, 1997*), and is often present at a ground cover of over 80% in peatlands (*Gunnarsson, 2005*).

During the last 25 years a significant proportion of blanket bog in the UK has undergone a programme of restoration. However, most of this work focused on degraded or eroded areas (mainly with grip blocking and/or re-vegetation) with little science-based evidence on the implications for ecosystem processes (*O'Brien, Labadz & Butcher, 2007*). Even less is known about the implications of current or potential alternative management options aimed at supporting 'active' bog plant communities, such as heather cutting (mowing), for the heather dominated seemingly 'intact' (i.e., carbon accumulating) blanket bog areas on many grouse moors on which this study focuses. However, as yet it remains unknown if mowing is an effective tool in reducing heather dominance (e.g., requiring likely repeated mowing). Although cutting is generally more expensive than burning, the sale of cut heather for commercial purposes (e.g., biomass for fuel or brash for bare peat restoration) can reduce the overall cost (*Backshall, Manley & Rebane, 2001*). Suitable machines include specifically designed heather flails and single or double-chop forage harvesters which can be attached to an all-terrain vehicle or tractor. The flail is normally set to between 12.5 cm and 15 cm above the ground. A clear advantage of cutting is that this activity is much less

constrained by weather conditions than burning (*Tucker, 2003*), although access for cutting machinery will be easier when conditions are drier and the ground is firmer (*MacDonald, 1996*). However, not all areas are accessible to large scale cutting or even less so bailing equipment (i.e., track width, remoteness and relief of the area and ground conditions such as very boggy or rocky areas).

A potential benefit of cutting (with leaving brash on site) is the reduced water loss via evaporation from the peat surface on brash covered ground compared to exposed burn areas, also limiting any negative rain/erosion effects as indicated by a post-management only plot level assessment by *Worrall, Rowson & Dixon (2013)*. Moreover, brash could effectively spread *Sphagnum* propagules across larger areas, thus facilitating restoration towards 'active' bog vegetation, which is the premise underlying the use of 'brashing' for restoration purposes in many areas of bare peat across the UK. As tractors are often used to manage large areas of land, a particular concern has been the potential impact of heavy machinery used to cut heather instead of burning it. In fact, cutting has not been recommended for blanket bog where regeneration is very slow and machinery could damage the vegetation (*DARDNI, 2011*). The two main concerns are (1) compaction and (2) reducing micro-topography by chopping off the tops of hummocks. Whilst compaction could cause a potential increase in bulk density reducing water storage (and reducing peat depth), a possible reduction of the micro-topography due to cutting off the tops of hummocks (e.g., protruding clumps of sedges) could impact surface water runoff (which is slowed down by a varied micro-topography) and also affect nesting habitat for ground nesting upland birds (such as Dunlin which prefer to nest within elevated hummocks).

This study compared grouse moor (i.e., heather) management impacts (specifically comparing burning versus cutting) on total peat depth, peat bulk density and peat surface micro-topography. Three grouse moor blanket bog sites in Northern England were studied, each consisting of two paired sub-catchments (burnt versus mown) with plot-level replicates for four main managements (uncut, burnt, mown with or without brash removal). Management impacts were compared against either pre-management measurements (i.e., all tall heather) or the uncut control plots (i.e., no management).

## MATERIALS & METHODS

### Field sites
The three study sites were three active grouse moor sites in north-west England. The names used to identify the sites throughout are Nidderdale, Mossdale and Whitendale (see Fig. 1). The sites were chosen based on a set of key criteria: all were classed as blanket bog with a mean peat depth of over 1 m (Histosol) classified in England and Wales as Winter Hill series (see *Cranfield University, 2018*). Typically, the sites were managed with a 10–15 year burn rotation (based on gamekeeper information) and all had a long history of burning (more than 100 years; based on estate information). All sites had more than 50% *Calluna vulgaris* (ling heather) cover, with at least some existing bog vegetation in the form of *Eriophorum* (cotton-grass; forming some hummocks) and *Sphagnum* moss species, and had a low sheep stocking density of <0.5 ewes ha$^{-1}$. The average climatic conditions across

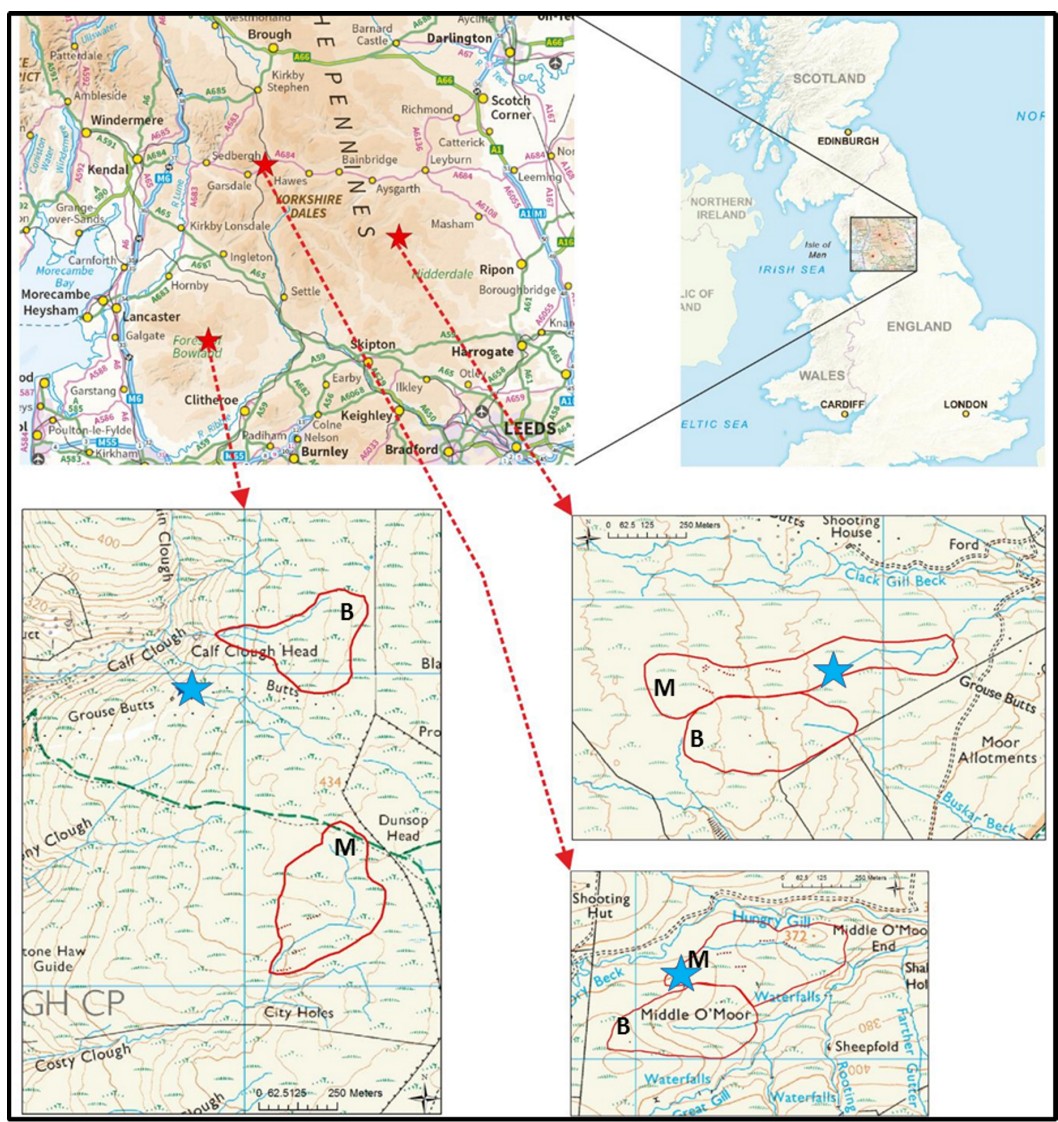

**Figure 1** **Field site and catchment locations to provide a geographic context.** Field site locations in north-west England (inset) in relation to the United Kingdom (outline). Shown are the three sites Nidderdale, Mossdale and Whitendale (indicated by the red stars). Maps downloaded: 09th September 2016 from MiniScale® [TIFF geospatial data] during download of GB tiles from Ordnance Survey (GB) using the EDINA Digimap Ordnance Survey Service (http://digimap.edina.ac.uk; ©Crown copyright and database rights (2016) Ordnance Survey (100025252). The catchment boundaries (thick red lines) with the burnt (B) and mown (M) catchments and automated weather station (blue star) are detailed in the lower maps for Whitendale, Mossdale and Nidderdale. Note the main stream within each sub-catchment and the contour lines.

the three sites over the period 2012–2016 (based on hourly weather station data; MiniMet, Skye Instruments Ltd, Llandrindod Wells, UK) recorded a mean annual temperature of 7.3 °C and a mean annual total rainfall of 1,800 mm. The three sites differed from each other, with Nidderdale in the driest, most degraded condition (with considerable *Hypnum*

**Table 1  Basic site information for the three blanket bog sites.** Basic site information for the three blanket bog sites Nidderdale, Mossdale and Whitendale. Altitude is given as metres above sea level (m a.s.l.). Climatic data are provided based on hourly weather station data from each site during 2012–2016 (see text for further details and standard deviations). Values in brackets indicate averages for burnt versus mown sub- catchments including uncut plots). Water table depth is negative (i.e., below the surface) and vegetation class was based on the NVC classification system and was determined by additional vegetation surveys (see *Morton, 2016*) for each management type at each site in 2012 (pre-management) using the MAVIS software (*DART Computing & Smart, 2014*). Whereas individual average annual water tables (based on monthly means) are shown, only an overall NVC category is shown as there was no difference between sites or managements.

| Site | Nidderdale | Mossdale | Whitendale |
|---|---|---|---|
| **Altitude (m a.s.l.)** | **450** | **390** | **410** |
| **Mean temperature (°C)** | **7.2** | **7.2** | **7.6** |
| 2012 | 6.6 | 6.7 | 7.2 |
| 2013 | 6.7 | 6.9 | 7.2 |
| 2014 | 7.9 | 8.0 | 8.3 |
| 2015 | 7.3 | 7.2 | 7.5 |
| 2016 | 7.3 | 7.3 | 7.8 |
| **Mean rainfall (mm)** | **1587** | **2029** | **1858** |
| 2012 | 2012 | 2405 | 2209 |
| 2013 | 1311 | 1708 | 1393 |
| 2014 | 1512 | 1943 | 1714 |
| 2015 | 1768 | 2437 | 2136 |
| 2016 | 1328 | 1685 | 1839 |
| **Mean water table (cm)** | **−12.5 (−13.2 vs −11.9)** | **−7.7 (−9.9 vs −7.2)** | **−9.0 (−8.7 vs −9.1)** |
| 2012 | −14.7 (−14.4 vs −12.3) | −5.2 (−8.2 vs −6.4) | −10.3 (−12.5 vs −10.8) |
| 2013 | −15.1 (−11.9 vs −11.5) | −7.3 (−10.5 vs −5.3) | −9.2 (−8.2 vs −8.7) |
| 2014 | −15.3 (−13.0 vs −12.4) | −9.5 (−10.9 vs −6.9) | −9.1 (−8.1 vs −9.3) |
| 2015 | −15.2 (−15.0 vs −13.5) | −7.5 (−10.0 vs −5.1) | −9.4 (−10.5 vs −9.8) |
| 2016 | −12.6 (−12.0 vs −9.8) | −10.2 (−9.6 vs −7.5) | −5.8 (−5.4 vs −8.7) |
| **Plot-level peat depth (m)** | **1.6 (1.8 vs 1.5)** | **1.2 (1.1 vs 1.3)** | **1.7 (1.6 vs 1.7)** |
| **Plot-level slope (°)** | **4 (3 vs 6)** | **6 (5 vs 7)** | **8 (6 vs 11)** |
| **Vegetation type; NVC class** | *Erica tetralix* sub-community of the *Calluna vulgaris –Eriophorum vaginatum* blanket mire community; **M19a** | | |

*jutlandicum* moss cover), Whitendale supporting a vegetation which is most uniformly of a 'typical bog' community though still dry and somewhat degraded, while Mossdale as the wettest site showed the largest *Sphagnum* moss cover, though this mainly reflected the presence of poor-fen vegetation characterised by *Sphagnum fallax*. Table 1 provides an overview summary for the basic site conditions in addition to the following individual site information (but also see Fig. 2 for site conditions and sedge hummocks):

**Nidderdale** is located on the Middlesmoor estate in upper Nidderdale, which lies within the Yorkshire Dales National Park, UK, at 54°10′07″N; 1°55′02″W (UK Grid Ref SE055747) about 450 m a.s.l. The site had a mean (±standard deviation) annual air temperature of 7.2 ± 0.5 °C and annual total precipitation of 1587 ± 211 mm, the mean annual water table depth was −12.5 ± 6.4 cm. The soil is a poorly draining organic peat (Winter Hill series) with an average depth of 1.6 ± 0.3 m across the experimental plots with an average slope of 4 ± 3° and peat depth across the catchments ranged from 0.2 m to 2.9 m. Most of the grips within the study area, which were dug about 40 years ago, were naturally infilled

**Figure 2 Site condition pictures to provide an upland blanket bog context.** Site conditions as obob-servedserved by ground-level pictures (credit A. Heinemeyer) taken in winter 2012 at each site for (A) Nidderdale, (B) Mossdale and (C) Whitendale. Note the burn areas with regrowing sedge cover (mostly cotton-grass (*Eriophorum* spp.)) on the otherwise heather (*Calluna*)-dominated blanket bog vegetation.

by 2010 and no further grip blocking took place during the study period. There were few gullies (similar to grips but naturally formed) at this site.

**Mossdale** is located in Upper Wensleydale within the Yorkshire Dales National Park at 54°19′01″N; 2°17′18″W (UK Grid Ref SD813913) about 390 m a.s.l. The mean (±standard deviation) annual air temperature was 7.2 ± 0.5 °C and annual total precipitation was 2029 ± 346 mm, the mean annual water table depth was −7.7 ± 5.7 cm. The soil is a poorly draining organic peat (Winter Hill series) with an average peat depth of 1.2 ± 0.4 m at the experimental plots with an average slope of 6 ± 3° and peat depth across the catchments ranged from 0.3 m to 2.1 m. Most of the grips within the study area, which were dug about 40 years ago, were naturally infilled by 2010. There were no gullies at this site.

**Whitendale** is located within the Forest of Bowland (an Area of Outstanding Natural Beauty; AONB), Lancashire, at 53°59′04″N; 2°30′03″W (UK Grid Ref SD672543) about 410 m a.s.l. The mean (±standard deviation) annual air temperature was 7.6 ± 0.5 °C and annual total precipitation was 1858 ± 308 mm during the five year study period, the mean annual water table depth was −9.0 ± 6.9 cm. The soil is a poorly draining organic peat in the Winter Hill series with an average peat depth of 1.7 ± 0.4 m at the experimental plots with an average slope of 8 ± 3° and peat depth across the entire catchment area ranged from 0.2 m to 4.5 m (i.e., with shallower areas on steep slopes). This study area had no grips, although several gullies were present in both catchments.

## Experimental design

Each site offered two adjacent (Nidderdale and Mossdale) or closely (ca. 1 km) located (Whitendale) sub-catchments at the same elevation (ca. 420 m) and of similar size (∼10 ha), with each allocated as either burning or cutting management. The entire manipulative experiment was based on a Before-After-Control-Impact (BACI) design (*Schwarz, 2015*; *Stuart-Oaten, Murdoch & Parker, 1986*), to enable robust statistical analysis of the *after* management effects in relation to the experimental treatment intervention compared to pre-existing differences *before* management (i.e., allowing comparison to observed differences or changes unrelated to management). Within each sub-catchment four blocks each with one 5 × 5 m plot-level replicate per management, were defined in March 2012 with at least 50 m between blocks and 5 m gaps between plots. Plots were marked out with

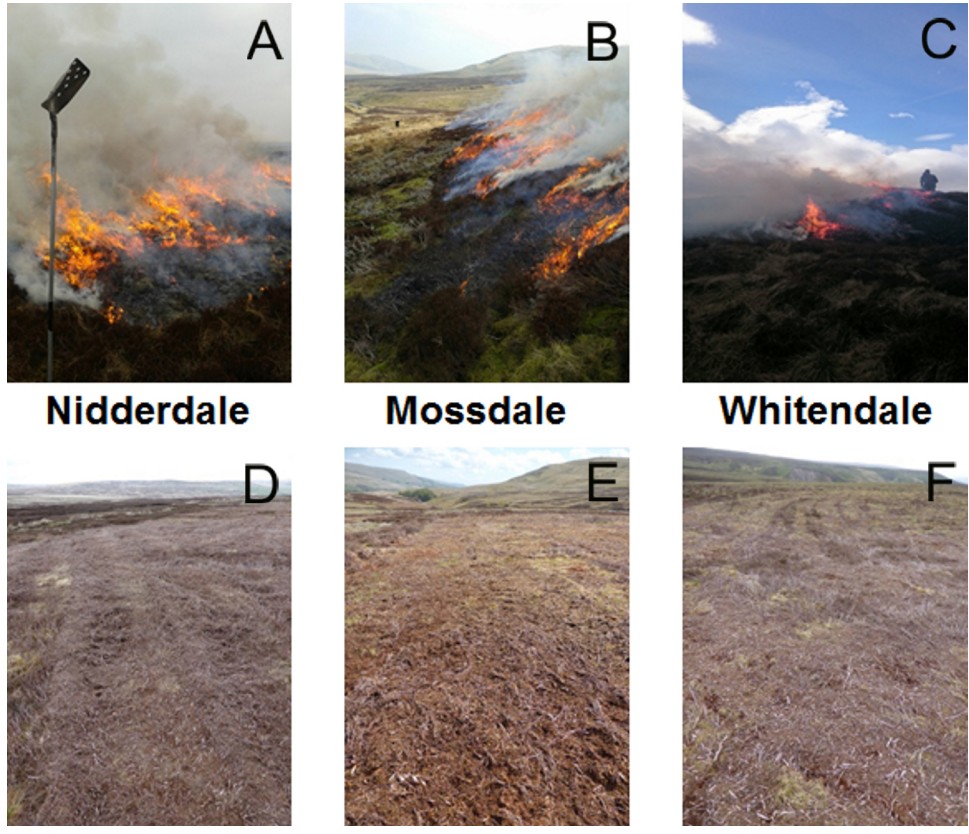

**Figure 3** **Site pictures providing a grouse moor management context.** The six sub-catchments (Nidderdale, Mossdale, Whitendale) shown during burning (A, B, C, respectively) and after mowing (D, E, F, respectively) in March/April 2013. For mowing machinery refer to Fig. 4.

wooden corner posts protruding approximately 50 cm from the peat surface. Water table depth was measured every 12 h (Omnilog, WT-HR 1000, TruTrack, New Zealand) with one dipwell per plot, which was always located at the lowest corner point of the plot (to capture hydrological impacts without the need for trampling over the plot). Burning and cutting (see Fig. 3 for example pictures of management at the three sites) were conducted as part of the usual management rotation of the grouse moors (aiming to manage heather areas every 10–15 years) with typical management areas of about 0.25 ha (50 × 50 m). Burning was done manually with gas torches setting alight the outside of the targeted heather area, whilst cutting was done using adapted tractors with double wheels (which at Mossdale were also fitted with caterpillar tracks to the back wheels) to reduce ground pressure to about 2 pounds per square inch (psi).

Figure 4 provides example pictures for the different mowing equipment used at the three sites. At Nidderdale a more basic and lighter cutting machinery (a small Case International 4,230 tractor, 85 horse power, with a back-fitted simple Bomford Topper (RS18) flail) was used, compared to the heavy machinery at Mossdale (a New Holland, 120 horse power,
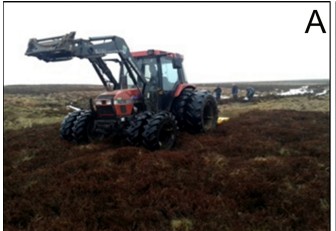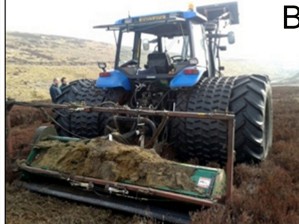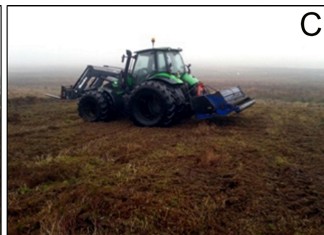

**Figure 4** **Site management pictures to provide a mowing context.** The three mowing arrangements: (A) Nidderdale, (B) Mossdale and (C) Whitendale. On average, vegetation was mown about 12 cm above the peat surface and the heather brash returned to the surface was about 5–10 cm long, with the coarsest brash at Nidderdale and the finest at Whitendale. The initial brash layer after mowing was around 5 cm thick.

**Table 2** **Description of the experimental design and sampling strategy.** Summary of the experimental design and sampling strategy across the three blanket bog sites. Numbers in brackets are the number of replicates or sampling strategies. Individual plot treatments had four replicates ($n = 4$), one in each of four blocks within each sub-catchment. See main text for further details.

| Sites (3) | | Nidderdale | Mossdale | Whitendale |
|---|---|---|---|---|
| Sub-catchments (2) | Burnt | | Mown | |
| Plot treatments (4) | Burnt (FI) | Uncut (DN) | Brash left (LB) | Brash removed (BR) |
| Sub-treatments (2) | – | – | *Sphagnum* pellet addition (+Sp) | |
| **Sampling strategies (2)** | | | | |
| Before & After | Bulk density (four depths: 0–5; 10–15; 20–25; 40–45 cm) Peat depth (as maximum: surface to bedrock) | | | |
| Control vs. Impact | Micro-topography (only in 2015) for five transects per $5 \times 5$ m plot (up to 45 points per plot) | | | |

with back-fitted Major hammer flails) and Whitendale (a Deutz Fahr tractor, 150 horse power, with a back-fitted heavy duty Ryetec flail mower).

Table 2 provides an overview for the experimental design and sampling strategy. In the burning sub-catchment (total of 4 plots per site), each block (4) contained one $5 \times 5$ m plot (FI) located within a burnt area (each ~0.25 ha). In the mown catchment (total of 20 plots per site), each block (4) contained five randomly allocated treatment plots (i.e., three main treatment plots and two more for manual *Sphagnum* pellet additions) located within a mown area (each ~0.25 ha); DN plots were left uncut as the 'do nothing' control, LB plots were mown with the brash left, BR plots were mown with the brash removed, LB+Sp plots were mown with the brash left and *Sphagnum* propagules added (this treatment was part of the overall project but plots are included in this analysis as they offered additional replication), BR+Sp plots were mown with brash removed and *Sphagnum* propagules added. Brash was removed from BR and BR+Sp plots by manual raking (~4–5 times after mowing; ca. 50 L brash were collected from the $5 \times 5$ m plots in 70 L bags and deposited in adjacent areas outside the experimental blocks. Experimental management started in 2013, with burning (Nidderdale: 5th March; Mossdale: 1st March; Whitendale: 21st February) and cutting (Nidderdale: 11th April; Mossdale: 9th April; Whitendale: 7th March) on all blocks.

On average, vegetation was mown about 12 cm above the peat surface (a standard height on heather-dominated grouse moors as per subcontractor and gamekeeper information) and the heather brash returned to the surface was about 5–10 cm long, with the coarsest brash at Nidderdale and the finest at Whitendale. However, the brash contained much finer pieces containing short shoot sections, leaves and moss fragments. The initial brash layer after cutting was around 5 cm thick.

## Micro-topography

The variation in the peat surface (micro-topography) was assessed on all 72 monitoring plots across the three sites in September 2015, two years after the onset of the experimental management, at Nidderdale on 14th, Mossdale on 15th and Whitendale on 16th. An initial visual assessment after management in early spring 2013 had revealed initial compaction of the peat surface during cutting (i.e., water table dipwells were pushed down by the tractors about 15–20 cm into the peat) which subsequently (on the same day) rebounded (visually detectable as the dipwell tube was left lower in the peat but the peat expanded again).

Two 60 cm canes were marked 20 cm from the bottom and notched 10 cm from the top. The canes were inserted into the peat at a right angle to the peat surface (slopes at plot locations were <5 degrees), exactly to the 20 cm mark, on either side of a 5 × 5 m plot. A piece of twine was tied tightly between the two canes so that it sat within the notches and was exactly 30 cm high either side of the plot. The offset between peat surface and the twine was measured at nine marked points (every 50 cm) along the 5 m section of twine spanning across the 5 × 5 m plot using a 1 m long wooden ruler (measured to the nearest 1 cm). The offset to the peat surface on the mown plots with brash left was measured by pushing the measuring rod through the brash layer (which was at most a few centimetres) to visually contact the peat surface. Transects spanned the plots (i.e., was across the slope as opposed to down it) and there were five transects per plot, with the first being 50 cm into the plot. This provided 45 points across each of the 5 × 5 m plots. However, in one mown plot (LB) at Nidderdale a section of 27 points could not be monitored as it was not mown due to a step in the topography, and in another mown plot (BR+Sp) five points were excluded as they were located over a ditch. Moreover, there were some missed recording points (across all plots) for Nidderdale (10) and Mossdale (1).

The mean offset and the standard deviation of the offsets were calculated for each plot, graphs were produced and both were tested with a two-way ANOVA (management and site as factors) in Microsoft Excel 2010 (Office 14).

## Peat depth

Peat depth was measured (to the nearest 0.1 cm with an estimated error of about 0.5 cm) manually on all plot locations across each site in July/August 2012, before any management change, and again in April 2013 shortly after the burning and cutting management. Sampling was done at the same location as the water table dipwells, where the initial compaction was observed immediately after mowing management. Before and after samples were obtained (peat depth) or taken (for bulk density see next section) from within one area of about 25 × 25 cm. The tractor path ran straight over this area (it was

guided so both sets of wheels, front and back, moved directly over the dipwell area, which therefore represented the maximum weight impact area). All locations were surveyed using commercial (Clarke CHT640, Clark-Drain, Peterborough, UK) 1.5 cm diameter PVC drainage rods (92 cm extendable sections with screw fittings) and peat depth was determined (excluding hummocks) by detecting a sudden resistance (i.e., hitting the bedrock/clay layer). A linear regression analysis was performed in Excel comparing pre-versus post peat depths for burnt and mown areas.

## Bulk density

As effects on surface compaction might not be detected in a measurement over the entire peat depth, an additional assessment of bulk density (BD) in the surface peat layers (0–50 cm) was undertaken and compared between pre- and post-management for the FI, DN and LB plots. For BD analysis a peat core was taken (see peat depth section) to 50 cm peat depth using a custom-made three-sided 5 cm square box corer of 1.1 m length with a cutting blade (avoiding compaction and producing very consistently shaped samples of reliable volume). Peat cores were taken twice, once in August 2012 (before management), and again in March 2014 (a year after management). The peat core was separated into individual sections for which 5 $cm^3$ samples were cut and placed in labelled storage bags. The sections sampled were: D1 (0–5 cm); D2 (10–15 cm); D3 (20–25 cm); D4 (40–45 cm).

BD was determined on the individual 5 $cm^3$ subsamples by first weighing individually labelled trays on a balance with a precision of 0.0001 g. The field wet peat samples were then removed from their storage bags and place in a labelled tray. The tray was then re-weighed to allow calculation of the (wet) sample weight. Foil trays containing samples were then placed in an oven at 105 °C to evaporate water from the samples. For extremely wet samples the oven door were left slightly ajar to let moisture escape (about 1–2 days was required). The samples were then dried completely with closed oven doors (which could take up to three days). Peat samples were dried until a constant weight was reached and stored in a desiccator until further analysis. BD was then calculated according to *Chambers, Beilman & Yu (2011)*.

## Statistical analysis

Regression fitting was performed in Excel (Microsoft Office v.14). All other statistics were performed in SPSS (v. 25). Micro-topography data was analysed as the standard variation (i.e., providing a measure of the magnitude or range of the plot-level micro-topography) around the mean offsets ($n = 4$ per management and site) by a two-way ANOVA (with site and management as main factors). Where significant differences were detected, the Tukey HSD test was used to determine between which groups significant differences occurred. A one-sample Kolgoromov Smirnov test was used to check whether the data followed a normal distribution and a Levene's test was used to assess homogeneity of variance. Data were $log_{10}$ transformed to fulfil the assumption of a normal distribution and fulfilled the homogeneity of variance assumption.

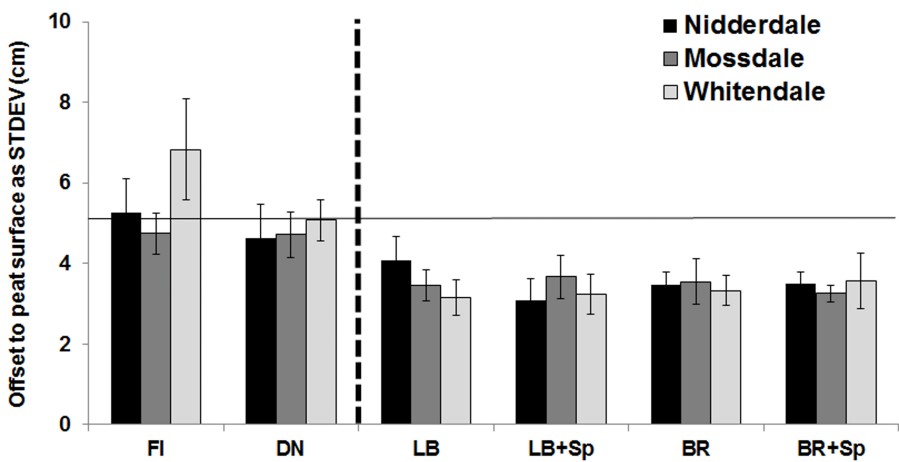

**Figure 5  Micro-topography measured as the standard deviation of the measured offset from the peat surface.** Mean standard deviation (STDEV $\pm$ SE) of the measured offset (in cm) in relation to the average peat surface outside the plot area obtained from the micro-topography monitoring across the $5 \times 5$ m plots (up to 45 points were recorded per plot, see methods) for each management ($n = 4$ for burnt (FI), uncut (DN), mown with (LB) or without brash (BR) and with or without *Sphagnum* pellet (+Sp) addition) for the Nidderdale, Mossdale and Whitendale blanket bog sites. The dashed black vertical line separates the burnt and uncut plots (left) from the mown plots (right), whilst the horizontal line indicates the average STDEV (5.22 cm) of the burnt and uncut plots across all sites.

# RESULTS

## Micro-topography assessment

The micro-topography (determined as the variation in peat surface across each of the $5 \times 5$ m plots across all management scenarios: burning, variations of cutting and uncut) showed that the mean offset variability (measured as the standard deviation of the measured offsets against the peat surface), which could be positive or negative (Fig. 5), was significantly ($p < 0.001$) higher on burnt and uncut plots (mean of 5.2 cm), and noticeably lower for all the mown plots at all sites (mean of 3.5 cm), but without any observable impact by brash removal or *Sphagnum* addition (both of which required walking across the plots and therefore could have also shown management related compaction). However, at Whitendale, the burnt plots had a higher variability in the offset to the peat surface than the uncut plots, the latter of which were located in the much flatter adjacent mown sub-catchment. Notably, the Whitendale burnt management area also had more gullies across the catchment (the cause for those gullies remains unknown).

The micro-topography impact was also clear when shown as the median and the interquartile ranges of the absolute offsets (Fig. 6). Apart from the burnt plots at Whitendale, all three sites had median offsets of close to zero for uncut and burnt plots, whereas mown plots had more negative offsets, with lower interquartile ranges, than uncut and burnt plots, particularly for Mossdale and Whitendale (apart from the offsets at burnt plots).

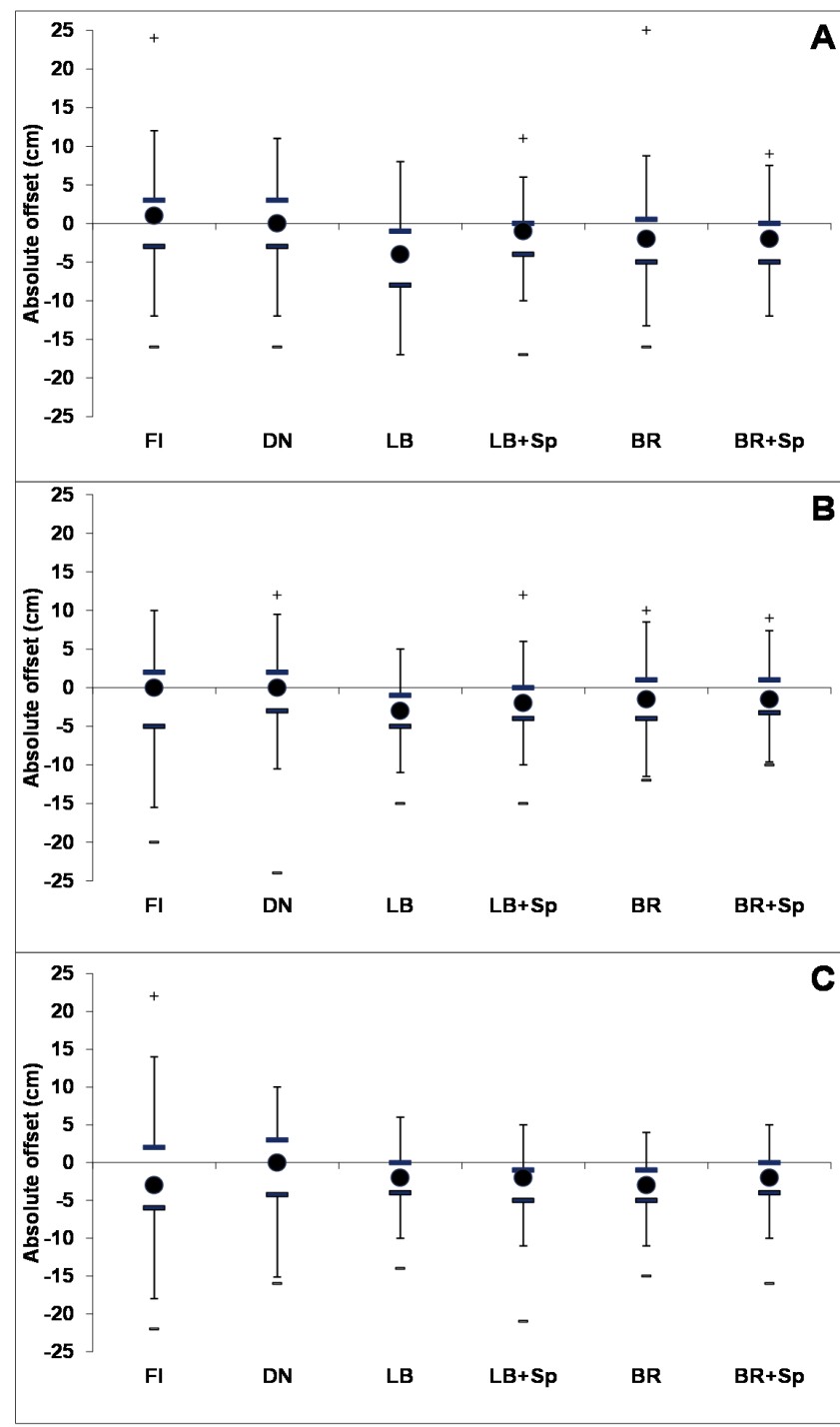

**Figure 6** **Micro-topography of the individual plot-level management treatments as the absolute offset in relation to the peat surface.** Mean absolute offset (in cm compared to the average peat surface (zero) outside the plot area) from the micro-topography point monitoring across the 5 × 5 m plots (up to 45 points were recorded per plot, see methods) for (A) Nidderdale, (continued on next page...)

**Figure 6 (…continued)**
(B) Mossdale and (C) Whitendale for each individual management ($n = 4$ for burnt (FI), uncut (DN), mown with (LB) or without brash (BR) and with or without *Sphagnum* pellet (+Sp) addition). Negative values relate to dips, positive values to hummocks. Outliers are shown as + and − with median values as filled circles, the thick lines indicate the interquartile range and the thin lines are the upper and lower confidence range (1.5 times the interquartile range).

## Peat depth assessment

Overall, peat depth did not show any significant differences between pre- and post-management measurements at any site, with the individual linear regression lines being very close to a 1:1 line (Fig. 7) for both burnt ($n = 4$) and all mown with or without brash removal ($n = 16$; i.e., excluding uncut) plots. However, Mossdale burnt and Whitendale mown plots showed a slight tendency for higher peat depths after than before management with increasing overall peat depth.

## Surface compaction assessment

The comparison of mean BD with standard errors (SE) showed higher BD after cutting management than before (Fig. 8), particularly in the two upper peat layers (0–15 cm), at Nidderdale (mean ± SE of $0.15 \pm 0.02$ vs $0.12 \pm 0.02$ g cm$^{-3}$, respectively) and Mossdale (mean ± SE of $0.09 \pm 0.01$ vs. $0.07 \pm 0.01$ g cm$^{-3}$, respectively). However, this 'compaction' effect was also observed for the burnt and uncut plots, both of which were not affected by the tractor. Moreover, Whitendale peat BD, unlike those at the other two sites, were remarkably similar between pre- and post-management assessments (mean ± SE values for the upper two peat layers on mown plots were $0.12 \pm 0.01$ g cm$^{-3}$ for both periods). Therefore, no further statistical analysis was performed as any significant differences would have indicated a compaction effect due to management although the burnt and uncut plots showed that this was clearly unrelated to cutting management but rather reflected a general fluctuation in BD and peat depth due to moisture changes as described by *Morton & Heinemeyer (2019)*.

## DISCUSSION

Currently, there is considerable UK government agency (i.e., Natural England) push to stop rotational heather burning on deep peat as part of grouse moor management and instead replace burning with alternative cutting but evidence on ecosystem physical, chemical and ecological impacts of both is still limited (*Harper et al., 2018*). This study has provided the first data on assessing potential peat physical and surface (micro-topography) implications of large scale cutting as part of alternative heather-dominated blanket bog management. The machinery used represented a common range of available equipment, including an estate owned small tractor with double wheels with a simple flail cutter (Nidderdale) and two larger sub-contractor mowers with a double chop cutter, one also with double wheels (Whitendale), the other with additional caterpillar tracks (Mossdale). The double wheel and track weight distribution is important as peat compaction would likely reduce peat water holding capacity and storage as seen in a review on the assessment of physical impacts on water flow and storage by *Rezanezhad et al. (2016)*. Such physical compaction impacts are

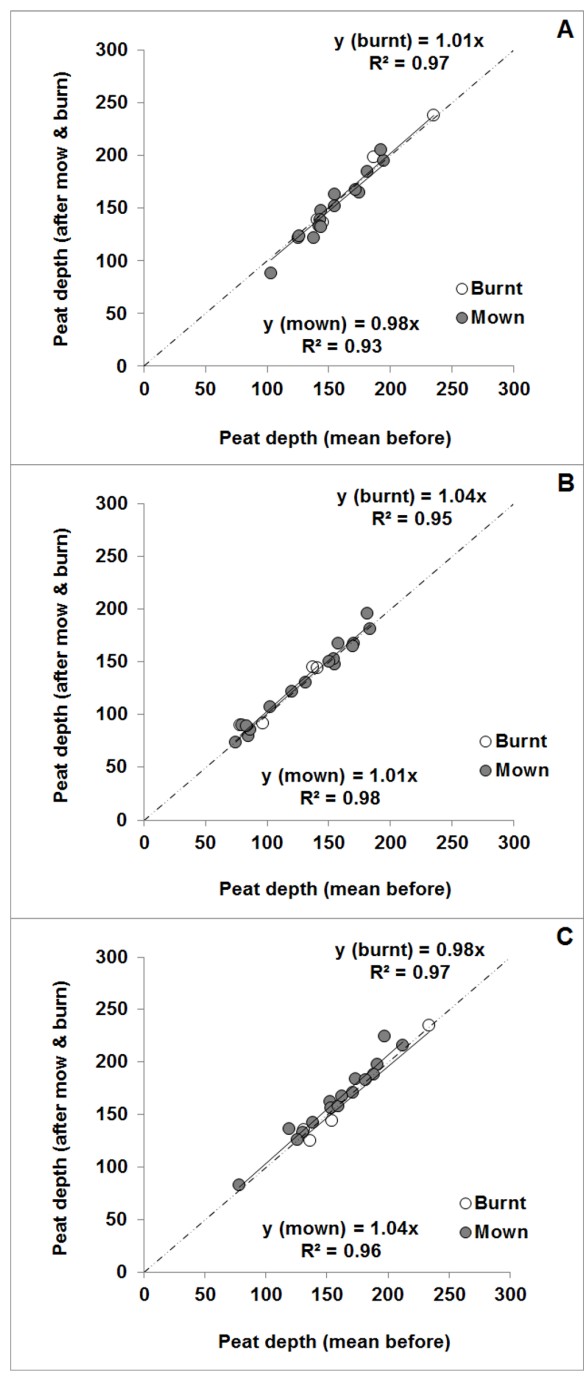

**Figure 7  Mean peat depth for the mown and burnt plot-level managements at the three sites comparing the after versus before management survey depths.** Mean peat depth (in cm) after plot-level management on mown (excluding uncut plots) and burnt plots plotted against mean peat depth before management for the three sites (A) Nidderdale, (B) Mossdale and (C) Whitendale. The linear regression equations and $R^2$ values are shown for each management (i.e., burnt vs. mown). All regressions were highly significant ($P < 0.001$) and were close to the 1:1 line (shown by a dashed line).

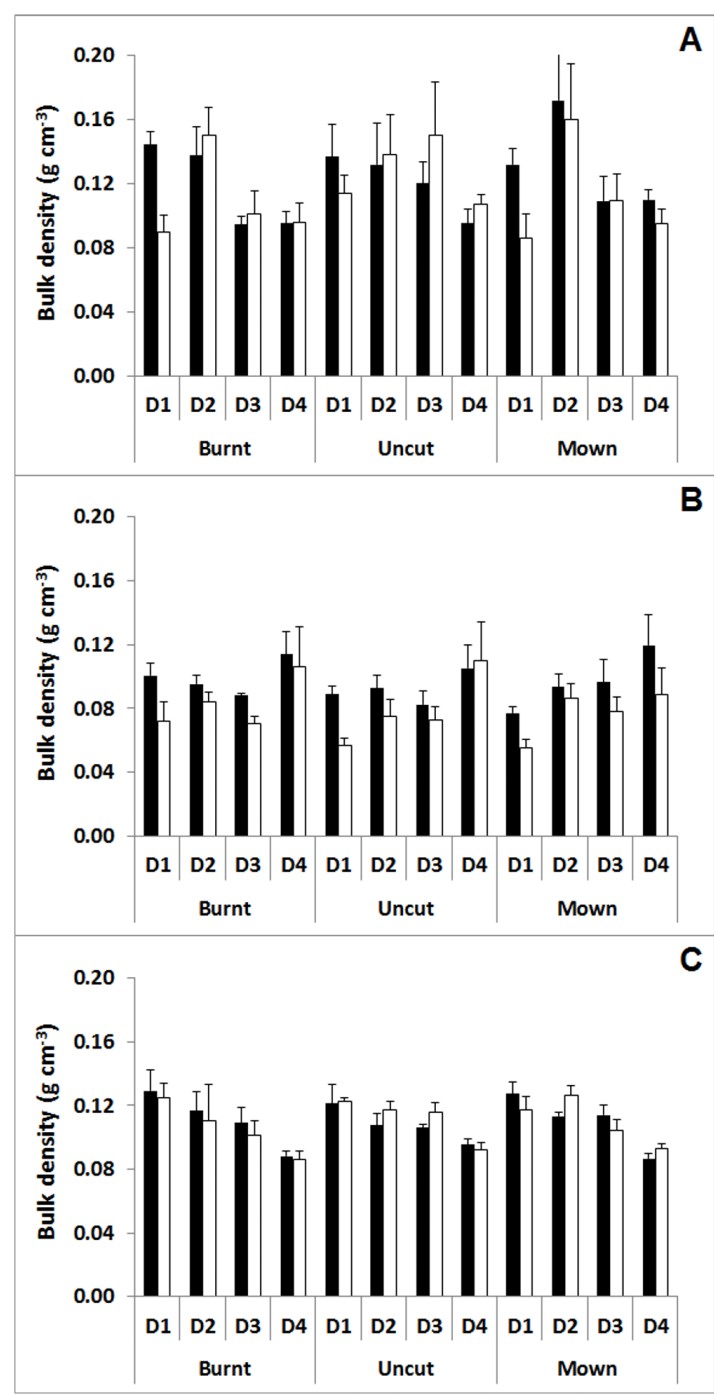

**Figure 8  Mean bulk densities for the burnt, uncut and mown (with brash left) plots at four soil depths comparing before versus after management survey data at the three study sites.** Mean bulk density (BD in g/cm³; shown with standard error; $n = 4$) measurements after (black bars) and before management (white bars) for burnt, uncut and mown (with brash left) plots at (A) Nidderdale, (B) Mossdale and (C) Whitendale. The four investigated BD peat depth layers were: D1 (0–5 cm); D2 (10–15 cm); D3 (20–25 cm); D4 (40–45 cm).

therefore likely to increase surface runoff via changes to the hydrological conductivity of the surface peat layer (*Rezanezhad et al., 2016*). Moreover, the cutting equipment is normally set high enough to avoid cutting hummocks and therefore reducing the micro-topography of the bog (but still needs to be low enough to cut heather efficiently), which has been shown to have important ecological via habitat diversity (e.g., for ground nesting birds such as the in the UK quite rare dunlin or golden plover: *Ratcliffe, 1990*) and hydrological (e.g., surface roughness and water flow/storage: *Aleina et al., 2015*; *Nungesser, 2003*) functions.

The micro-topography assessment did not include a pre-management change measurement. However, the before-after comparison was provided by comparing the managed plots to the uncut (in this instance the control) plots, which were located within the mown area but were not impacted on by the machinery. This comparison provided strong evidence that plot micro-topography was made less variable, as seen in the lower offset variability on mown plots, through the cutting off the tops of the sedge hummocks (Fig. 5). Cutting impacts were slightly less at Nidderdale (when comparing the LB plots), where more basic and lighter cutting machinery (a small tractor with a simple flail) was used, compared to the heavy duty tractors with hammer flails and a fine double chop at Mossdale and Whitendale. Consequently, the largest impact was observed in all mown treatments (across all mown plots compared to burnt and uncut) for Mossdale, where the median of the absolute offset (Fig. 6) was lowest, and for Whitendale, where the interquartile range (Fig. 6) was lowest. Whilst there were some differences in peat depth and also vegetation cover between sites and burnt and mown catchment plots, it is important to note that there are two comparisons to assess mowing impacts: (1) mown plots vs. burnt plots (with the caveat of slightly different conditions such as in peat depth, see Fig. 7, yet identical NVC classification) and (2) mown vs. uncut plots (with very similar conditions as placed next to each other in each block).

The assessment of peat depth comparing before *versus* after management measurements did not reveal any meaningful or clear differences in relation to compaction due to cutting (Fig. 7). However, relatively small changes in surface peat depth could remain hidden within even a small peat depth measurement error (which is estimated to be less than 5 cm). Moreover, the observed peat compression during cutting (of about 15 cm, see methods) indicated the potential for a lasting BD impact in the surface peat layer. Therefore the top 50 cm were investigated further by detailed BD assessments.

The observed higher BD at Nidderdale and Mossdale on mown plots (Fig. 8) might indicate compaction by cutting machinery. However, this was the case across all managements including the burnt and uncut plots, despite no tractors entering either the burnt or uncut plot areas. Therefore, the observed change in BD was evidently unrelated to management and reflected a natural process of BD change. A well-known natural process is "bog breathing" (*Ingram, 1983*) which causes peat level changes due to shrinkage and expansion in relation to water table and moisture changes. That this underlying natural cause explained the observed change in BD was supported by a larger apparent increase in BD at the surface for both Nidderdale and Mossdale (Fig. 8), which could be related to differences in rainfall amounts and thus peat moisture. Whereas pre-management samples were taken after a particularly wet summer period in 2012, leading to peat expansion and

thus lower BD, post-management samples were taken after a particularly dry spring in 2014 (Heinemeyer et al., unpublished climate data), leading to peat shrinkage and thus higher BD as shown for the same study sites by *Morton & Heinemeyer (2019)*. Moreover, the remarkably similar BD at Whitendale (between pre- and post-management assessments) could be linked to likely similar moisture conditions in relation to high rainfall events (~60 mm) in mid-March 2014 (two weeks before post-management sampling), and a further ~30 mm of rain during the week of sampling (all Heinemeyer et al., unpublished climate data), causing peat expansion and thus a decrease in BD.

Notwithstanding the lack of evidence on cutting causing possible peat depth or peat compaction issues, this plot-level assessment was of low replication and did not consider possible impacts elsewhere in the managed catchment area, such as stopping and turning points of the heavy machinery. Moreover, this study only assessed plot-level impacts from a one-off management; when heavy machinery is to be used, particularly if access is frequent, track impacts would need to be addressed across the wider landscape area. Future work should assess such potential cutting impact areas, ideally also considering a broader range of blanket bog conditions (i.e., impacts might be more severe on wetter, softer and more hummock containing bogs).

## CONCLUSIONS

Despite the expected and observed initial compaction of the peat surface after cutting with heavy machinery (i.e., tractors), there was no lasting effect of mowing management treatment on either peat depth or bulk density. However, plot micro-topography was reduced after cutting by chopping off the tops of some and mostly sedge-dominated hummocks. Further research might consider wider landscape assessment of cutting equipment, particularly considering points where machinery stopped and turned.

## ACKNOWLEDGEMENTS

We would like to thank Anda Baumerte for assistance during the field work (micro-topography assessments). Thanks also to everyone involved in providing access to the three sites (specifically the land owners supporting the science and the gamekeepers putting up with scientists roaming the land in search for answers which seem far removed from red grouse).

### Funding

Partial funding for this project was granted by the Department for Environment and Rural Affairs (Defra) UK (BD5104). The funders had no role in study design, data collection and analysis, decision to publish, or preparation of the manuscript.

### Grant Disclosures

The following grant information was disclosed by the authors:
Department for Environment and Rural Affairs (Defra) UK: BD5104.

## Competing Interests

The authors declare there are no competing interests.

## Author Contributions

- Andreas Heinemeyer conceived and designed the experiments, performed the experiments, analyzed the data, contributed reagents/materials/analysis tools, prepared figures and/or tables, authored or reviewed drafts of the paper, approved the final draft.
- Rebecca Berry performed the experiments, analyzed the data, reviewed drafts of the paper.
- Thomas J. Sloan performed the experiments, authored or reviewed drafts of the paper.

## Data Availability

The raw measurements are available as File S1 (bulk density) and File S2 (micro-topography). The raw data includes all plot-level measurements based on peat cores (S1) and field measurements of peat surface offsets (S2).

## Supplemental Information

Supplemental information for this article can be found online at http://dx.doi.org/10.7717/peerj.7298#supplemental-information.

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
