# Peer review of "Assessing soil compaction and micro-topography impacts of alternative heather cutting as compared to burning as part of grouse moor management on blanket bog"

_PeerJ, doi:10.7717/peerj.7298_

## Round 0.1 · original submission · Major Revisions

Dear Andreas and Thomas

After reading the comments of the reviewer´s I think that this manuscript would need a very thorough revision before being publishable in PeerJ.
Besides some important methodological concerns (see reviewer 1 comments), there are also some conclusions drawn by the author´s that will need either to be better supported, or if not, to be modulated and/or re-formulated.

Please read along reviewer´s comments and address them all in a future version.

Reviewer 1 ·

Basic reporting

Comments to the author Manuscript Number: 29086
Comments to: “Assessing soil compaction and micro-topography impacts of alternative heather cutting as compared to burning as part of grouse moor management on blanket bog”
The study addresses an interesting issue related to how avoid compaction in organic soils. The theoretical basis of this paper looks solid. However, I recommend a major revision of the manuscript.

Experimental design

The experimental design is so complex! Your data could be represented by four factors:
- 5 Treatments (FI, DN, LB, BR, LB+Sp, BR+Sp)
- 3 Sites (N, M, W)
- 4 Sampling Depths
- 2 Before-After control impact
It is very difficult to understand the results. I suggest you:
1. Add a picture of your experimental design for each of your figures (Probably, it will be supplementary files)
2. Determine the statistical analysis for each experimental design, e.g. three-way ANOVA, two-way ANOVA, one-way ANOVA, Kruskal-Wallis, etc, indicating how did you validated your analysis (data normality or the equality of error variances (Levene’s test))
3. Indicate how did you validated your regression analysis (outlier presence, normality of unstandarised residuals, etc)
4. Please explain what kind of software was used.

Validity of the findings

Other comments are:
Introduction
Lines 118-120: Please add a reference to support this sentence.
Lines 121-123: Please add a reference to support this sentence.
Lines 136-139: Please define clearly the main objectives of your study. This phrase is too ambiguous.
Lines 159-183: I suggest you to do summary this information.
Line 193: What are the treatments?
Lines 216-219: Please explain the meaning of DN, LB, BR, etc, in the main text, not just in the legend of figures.
Line 288: I suggest you to add a Statistical analysis section, explaining it.

Results and discussion section need to be modified, in agree with the changes suggested.

Additional comments

-

·

Basic reporting

The manuscript presented is generally well written and the language is professional, but currently the introduction promotes cutting as a means of reducing the dominance of heather where there is no evidence. The text does mention that there is a lack of evidence on the implications of cutting but the current benefit of cutting promoted here is speculative.
Heather dominated blanket bog is degraded and rather than continuing to promote dominance of Calluna these ecosystems need rewetting to promote diversity of floral/bryophyte assemblages and reduce the dominance of Calluna. In lines 97-100 the authors state:
“Moreover, alternative heather management methods could also be used to reduce ‘over’-domination of heather in favour of more diverse and ‘active’ peatland forming vegetation, especially with regard to the restoration of peat hydrological function and Sphagnum growth.”
This statement needs supporting with evidence or must be recognised as speculation. I am not aware of any published evidence that demonstrates that cutting vegetation reduces heather dominance. This is important context here because grouse moor management aims to retain a dominance of heather, so it is unclear why a management technique that might reduce heather would be supported by land managers. It is key that the authors highlight to the reader that heather cutting might still promote heather dominance as publications such as this could be used by a range of stakeholders and feed into future management strategies. In lines 128-129 the authors refer to a report by DARDNI (2011) that states cutting should not be used on blanket bog as the machinery could damage the vegetation. It is surprising therefore that no assessment of vegetation was made in this study. This is crucial and indeed the author submission states:
“Notably, peat could compact under the weight, thus affecting eco-hydrological processes and ultimately carbon cycling and storage, but could also affect water cycling by altering micro-topography, vegetation composition and thus runoff rates.”
The wider impact of cutting machinery on blanket bog must also be highlighted in a publication such as this. Adoption of cutting will require repeatedly getting heavy machinery to the area marked for cutting but in so doing will undoubtedly affect surrounding areas. This will likely involve the use of tracks or routes which damage blanket bog (see e.g. Lindsay et al., 2016; IUCN briefing note 12) and continued use of the same route may alter the hydrology, perhaps even create new surface flow and promote drainage. If managing a central area of blanket bog by cutting results in drainage of the edges of the bog then cutting may not be an appropriate management technique. This must be highlighted in the introduction and discussion.
In addition, the authors note that ‘rebound’ occurs (line 240) when a machine moved over the peat, but the damage is far more significant where the vehicle stops and/or turns. It is of concern therefore that the authors did not assess this (line 387) although they do point out that it needs to be assessed in future (line 389). The manuscript must address the wider impact of cutting machinery.
Figures
It is very difficult to understand the study sites and experimental design from Figures 2-3. The experimental design text from line 185 onwards could neatly be supported with a schematic diagram and additionally an overview figure that presents the sites in aerial photography and the location of the plots with contour lines (or some presentation of elevation to show any variation in topography) would be extremely informative. Grips and gullies are mentioned and the locations of all these features need to be presented – these features will affect the data and need to be included in analysis and interpretation.
Figure 3 shows some heather being burned but with no context. The burning on Mossdale appears to be on an area of relatively high slope, something that is uncharacteristic of blanket bog. Is this area one of the experimental plots? Line 243-244 states “slopes at plot locations were <5 degrees”. Also line 251 states that there was a “step in topography”. These observations raise questions as to whether all plots are comparable topographically.
For Figure 3 it might be more informative to show the same plot before and after burning or mowing.
Raw data
The data are presented clearly and well annotated though there is a minor inconsistent labelling (the microtopography spreadsheet uses ‘M’ where graphs, text and other data spreadsheets report ‘LB’)

Experimental design

The design of the experiment provides a good number of treatments and replications with which statistically robust analysis could be undertaken. However, there are a number of serious concerns about the data and analyses presented.
Different catchments
At all sites the burn treatment is in a different catchment to all the mow treatments and it must be demonstrated that peat depth, topography (both micro- and meso-) and vegetation is comparable between catchments prior to the experimental manipulation. These data may exist and ought to be presented.
Microtopography
The approach of measuring surface variation at 45 points in each plot appears to provide a good number of data for analysis. However, the approach does not provide a before and after data for comparison – the authors note on line 351 that “The micro-topography assessment did not include a pre-management change measurement”. This would be the most robust measurement for comparison of change.
The use of data from uncut plots may provide some insight, but as noted above with the catchment descriptions, a full description of all plots needs to be provided to demonstrate that the uncut plots are comparable to the manipulated plots prior to the experiment. It is crucial that the microtopography is described as erosion features, ditches, and gullies are mentioned in the site descriptions but not accounted for in analysis. A full vegetation survey including indicators such as the particular species of Sphagnum present is also needed as this provides insight to the microtopography. Figure 5 shows significant variation in topography between the mown sites with brash where Sphagnum was and was not applied. It is unlikely that the addition of Sphagnum propagules would have any affect and suggests that the plots were not comparable prior to the experiment. This indicates that the variation recorded between replications cannot be interpreted without actual pre-treatment measurements.
Can you also explain how the microtopography of the mown plots with brash left was measured? Was the brash removed prior to measurement? Or is the brash included? What impact does this have on the data?
Peat depth and bulk density
There is no description in the method of precisely where samples were taken/measured. For both these measurements it needs to be demonstrated that the ground where samples were measured/collected were in fact impacted by the machine(s). There is a gap between the wheels (under the tractor). Does the mower have a roller that runs along the surface? Some areas will be impacted by both front and rear tyres, while some will be impacted by just rear tyres. There needs to be a lot more detail in the description of the footprint of the mower and tractor and a clear description of where samples were taken and why.
Ideally the experiment needs to collect measurements from all combinations of machine impact as this will vary and might provide insight as to the optimal vehicle setup to cause the least damage. See also note above about not assessing where vehicles stop/turn.
Peat depth
Unless I have misunderstood, one peat depth measurement was taken in each plot. Peat depth varies spatially over very short distances, often following changes in the underlying substrate, and therefore significantly more than one measurement of depth is needed to provide a representative record for an area of 25 m2. The mean peat depth data do not provide a comparison of before and after, they simply provide two sets of randomly located peat measurements. No interpretation or conclusions can be drawn from these data. If RTK DGNSS was used to take measurements at exactly (+/- 2 cm) the same location this needs to be stated as it may increase the validity of the observations.
Bulk density
In Figure 8, the data for Mossdale show that for 11 of the 12 samples BD increased after management intervention including in uncut areas. The authors note on lines 371-373 “Therefore, the observed change in BD was evidently unrelated to management and reflected a natural process of BD change.”
Lines 374-385 indicate that no interpretation can be drawn from these data: “That this underlying natural cause explained the observed change in BD was supported by a larger apparent increase in BD at the surface for both Nidderdale and Mossdale (Figure 8), which could be related to rainfall and thus peat moisture. Whereas pre-management samples were taken after a particularly wet summer period in 2012, leading to expansion and thus lower peat bulk density, post-management samples were taken after a particularly dry spring in 2014 (Heinemeyer et al., unpublished climate data), leading to peat shrinkage and thus higher BD. Moreover, the remarkably similar BD at Whitendale (between pre- and post-management assessments) could be linked to similar moisture conditions in relation to high rainfall events (~60 mm) in mid-March 2014 (two weeks before post-management sampling), and a further ~30 mm of rain during the week of sampling (all Heinemeyer et al., unpublished climate data), causing peat expansion and thus a decrease in BD.”
In order to interpret these data there must be some inclusion and analysis of moisture content. If soil moisture was not recorded in the field, it might be possible to estimate PET and net moisture from temperature and rainfall data. This could potentially be used to quantify/model the change in moisture to enable comparison of the data.

Validity of the findings

The conclusions drawn are not supported by the data or analyses presented.
For peat depth there are insufficient data and for BD, in lines 371-373 the authors state;
“Therefore, the observed change in BD was evidently unrelated to management and reflected a natural process of BD change.”
It is therefore not possible for the authors to conclude that:
“…there was no lasting effect of management treatment on either peat depth or bulk density” (lines 394-395). Natural variation as a result of changing moisture needs quantifying.
There may be potential in the microtopography data – see comments in previous section.
It may be that more data exist which can be used to undertake major revisions of the analyses presented here.

Additional comments

My comments may overlap slightly across the review sections but I hope that the comments are clear and constructive. The study is timely as burning of vegetation on blanket peat/bog is being phased out and cutting does appear to be being adopted as an alternative management strategy. This study therefore has the potential to provide some crucial insight into the impacts of cutting on peat compaction and microtopography. The overall design of the experimental plots appears to provide a good number of treatments and replicates, however the conclusions regarding the impact on peat depth and bulk density are not supported by the data or analysis presented. It may be that more data exist with which to add confidence to the analysis and therefore need including to allow conclusions to be drawn.
There is potential in the microtopography data but far greater detail is needed. There are no data presenting the vegetation or microtopography in the plots. These data are needed to demonstrate that the plots were comparable prior to experimental manipulation. The site description simply mentions Sphagnum species – it is important here to identify the species as they occupy different microtopographic zones. Microtopography is more than hummocks and hollows. In line 133 the authors state “hummocks (e.g. protruding clumps of sedges)…” – this sounds like a description of a ‘tussock’ not a hummock. Revegetating erosion features may appear to be ‘hollows’ but the vegetation is required to classify the microtopography and identify whether it is a hollow or an erosion feature. Erosion features may revegetate with Sphagnum spp. but the presence of species such as deer grass (Trichophorum cespitosum) are clear indicators that this would not be a hollow. Without this information it is not possible to understand if the sites or plots are comparable.
The wider impact of cutting machinery on blanket bog must also be highlighted and the manuscript requires major revisions prior to consideration for acceptance. As these revisions relate primarily to the data and analyses I have not commented as much on the written component.

---

## Round 0.2 · Minor Revisions

Thanks a lot to the reviewers for their critical view and the author´s for their thorough revision. While I think the quality of the MS has been substantially improved with the corrections, there is still a number of issues that needs clarification. I agree with the reviewer that there should be highlighted somewhere that one of the problems of this study could be he low number of samples, perhaps not as representative of the heterogeneity as should be, given the well-known spatial heterogeneity of soils. This is important, since this increase in the BD that author´s explains as a natural process is one of the key finding of this study. The reviewer raise further concerns on how microtopography was assessed, providing some solutions to resolve it.

Please address these and further concerns from the reviewer, specially concerning the methodology, in a future version of this MS.

·

Basic reporting

As this is a review of a revised manuscript I have submitted a separate document containing my comments

Experimental design

As this is a review of a revised manuscript I have submitted a separate document containing my comments

Validity of the findings

As this is a review of a revised manuscript I have submitted a separate document containing my comments

Additional comments

As this is a review of a revised manuscript I have submitted a separate document containing my comments

---

## Round 0.3 · accepted · Accept

Author´s have satisfactorily answer to all the reviewer´s concerns so I am glad to accept this manuscript for publication in PeerJ.